# Modelling primaquine-induced haemolysis in G6PD deficiency

**James Watson[1,2]\*, Walter RJ Taylor[1,2], Didier Menard[3], Sim Kheng[4], Nicholas J White[1,2]**

[1]Mahidol Oxford Tropical Medicine Research Unit, Faculty of Tropical Medicine, Mahidol University, Bangkok, Thailand; [2]Centre for Tropical Medicine and Global Health, Nuffield Department of Medicine, University of Oxford, Oxford, United Kingdom; [3]Unité d'Epidémiologie Moléculaire du Paludisme, Institut Pasteur du Cambodge, Phnom Penh, Cambodia; [4]National Center for Parasitology, Entomology and Malaria Control, Phnom Penh, Cambodia

**Abstract** Primaquine is the only drug available to prevent relapse in vivax malaria. The main adverse effect of primaquine is erythrocyte age and dose-dependent acute haemolytic anaemia in individuals with glucose-6-phosphate dehydrogenase deficiency (G6PDd). As testing for G6PDd is often unavailable, this limits the use of primaquine for radical cure. A compartmental model of the dynamics of red blood cell production and destruction was designed to characterise primaquine-induced haemolysis using a holistic Bayesian analysis of all published data and was used to predict a safer alternative to the currently recommended once weekly 0.75 mg/kg regimen for G6PDd. The model suggests that a step-wise increase in daily administered primaquine dose would be relatively safe in G6PDd. If this is confirmed, then were this regimen to be recommended for radical cure patients would not require testing for G6PDd in areas where G6PDd Viangchan or milder variants are prevalent.

\*For correspondence: jwatowatson@gmail.com

**Competing interests:** The authors declare that no competing interests exist.

## Introduction

### Radical cure of vivax malaria in G6PD deficient patients

*Plasmodium vivax* accounts for over half the world's malaria burden outside sub-Saharan Africa (*Gething et al., 2012*). The control and elimination of vivax malaria require both cure of the blood stage infection (the stage that causes acute illness) and the prevention of later relapses which derive from dormant hypnozoites in the liver (radical cure). Hypnozoites are formed from sporozoites, which do not develop immediately following mosquito inoculation but instead remain dormant in hepatocytes for weeks or months before developing and causing recurrent blood stage infections called relapses. In general, *P. vivax* infections in tropical regions are associated with frequent relapses (with intervals as short as three weeks) whilst relapses in *P. vivax* infections from Central America, Northern India and temperate regions are associated with longer intervals from acute infection to first relapse (*White, 2011*).

Primaquine, an 8-aminoquinoline, is currently the only widely available antimalarial drug for the radical cure of *P. vivax* infections. Primaquine causes predictable oxidant haemolysis in G6PD deficiency (G6PDd) one of the most common genetic abnormalities of man (*Cappellini and Fiorelli, 2008*). Throughout Asia, the Mediterranean littoral and Africa, allele frequencies for this enzyme deficiency vary between 3% and 35% in the populations at risk from vivax malaria (*Howes et al., 2013*). As G6PDd has sex-linked inheritance, males are either deficient (hemizygotes) or normal, whereas women can be deficient (homozygotes), normal or partially deficient (heterozygotes) in

**eLife digest** Malaria is the most important parasitic disease that affects humans. Over half of the malaria cases in Asia and South America are caused by a species of malaria parasite called *Plasmodium vivax* (known as vivax malaria). This form of malaria results in repeated illness because dormant parasites in the liver wake at intervals to infect the blood. The only available drug that can stop these relapses is a drug called primaquine, which was developed seventy years ago.

Unfortunately, primaquine causes dangerous side effects in certain individuals who are deficient in an enzyme called G6PD, which helps defend red blood cells against stresses. Primaquine damages these cells so that they burst, leading to anaemia. This is a major problem because G6PD deficiency is common in regions where malaria is present: in some areas up to 30% of the population may be G6PD deficient. Since G6PD testing is not widely available, doctors often avoid prescribing primaquine to treat malaria, which results in more cases of disease relapse. Failing to prevent vivax relapses causes extensive illness and hinders efforts to eliminate malaria.

Is there a way to give this drug to patients that would be safer for people with G6PD deficiency? Primaquine destroys older rather than younger red blood cells. Watson et al. used mathematical modelling to see whether it is possible to develop a primaquine treatment strategy that would allow a gradual destruction of older red blood cells in individuals with G6PD deficiency, which would be safer. The mathematical model incorporates data from previous studies in malaria patients and healthy volunteers with G6PD deficiency and combines this with knowledge of how red blood cells are produced and destroyed. Watson et al. predicted that giving primaquine over 20 days in a steadily increasing dose was safer than current recommendations.

Mathematical models are simplifications of real world processes. The only way to test these findings properly will be to run a clinical trial that gives healthy volunteers who are G6PD deficient a course of primaquine treatment with a steadily increasing dose.

proportions determined by the Hardy-Weinberg equilibrium. Because of Lyonisation, there is substantial variability in the proportion of red cells which are deficient in individual heterozygote females (*Beutler et al., 1962*).

The degree of haemolysis following primaquine depends on the dose administered and the severity of the enzyme deficiency (and in heterozygote females the proportion of erythrocytes which are deficient). The more severe G6PDd variants found in South East (SE) Asia (for example, Viangchan, Mahidol, Coimbra, Union) and the Middle East/West Asia (for example, Mediterranean) are generally associated with more severe haemolysis compared to the common African A- variant. For G6PD normal patients, the primaquine regimen for radical cure that is recommended in SE Asia and Oceania (where relapse rates are high) is 0.5 mg base/kg/day for 14 days. Elsewhere it is 0.25 mg/kg/day for 14 days. For patients with G6PDd, a weekly dose is recommended; 0.75 mg/kg/week given for a total of 8 doses. Unfortunately G6PDd testing is not widely available despite the recent introduction of point-of-care rapid diagnostic tests (RDTs) for G6PDd. These RDTs are currently too expensive to deploy on a wide scale and can be difficult to interpret, and thus are not generally available (*Brito et al., 2016-08*; *Satyagraha et al., 2016*; *Oo et al., 2016*). Thus, primaquine is commonly not given to patients to avoid the risk of haemolysis so the burden of vivax malaria remains high, causing considerable morbidity and economic loss (*Price et al., 2007*).

## Mechanisms of red blood cell production

The mechanisms regulating red blood cell production and turnover have been well characterised. Red blood cells (RBCs) transport oxygen which is reversibly bound to the main red cell protein, haemoglobin. RBC production in the bone marrow is regulated to maintain oxygen carrying capacity. When the haemoglobin concentration in the blood falls, this reduces oxygen carriage and RBC production is up-regulated, a process mediated largely by the renal hormone, erythropoietin. At times of increased bone marrow production, reticulocytes appear in increased numbers in the circulation (the upper limit of normal is $\approx 1.5\%$). Normal RBCs in healthy people have a very stable life expectancy of around 120 days. This is well modelled by a Gumbel distribution with low variance. In

nucleated cells G6PD can be newly synthesised, but the red cells lose their nucleus before leaving the bone marrow so very young red cells (reticulocytes) have the highest G6PD activity, and this declines as the RBCs age. In most G6PDd variants, the mutant enzyme degrades more rapidly compared to the normal enzyme. Older erythrocytes may have up to five times less G6PD activity than reticulocytes. G6PDd results in lowered NADPH and a reduced ability to regenerate reduced glutathione. Reduced glutathione protects normal RBCs against oxidant stresses such as the haemolytic effects of primaquine metabolites and certain foods, classically fava beans. G6PD is also important for the function of catalase, another oxidant defence mechanism. As these non-reusable oxidant defence reserves are 'used up', the aging erythrocyte becomes increasingly vulnerable to oxidant haemolysis (*Beutler et al., 1954a*; *Dern et al., 1954*; *Beutler, 2008*; *Recht et al., 2014*).

## Evidence from previous studies of oxidant haemolysis in G6PD deficiency

As young red cells have more functional enzyme than older cells, the degree of oxidant haemolysis depends on the genetic variant of G6PDd and the age distribution of the red cell population. Once the older cells have haemolysed, the remaining younger erythrocytes are essentially resistant to further damage by the same dosing regimen (that is, drug exposure) (*Beutler et al., 1954a*). However, higher primaquine doses do cause further haemolysis. This explains the fall then rise in haemoglobin with continued daily primaquine administration in mild and moderate severity variants of G6PDd. This temporary primaquine insensitivity in G6PDd individuals with the continued primaquine administration was characterised by Beutler and colleagues in a series of studies conducted over sixty years ago (*Beutler et al., 1954a,1954b*, *1955*; *Dern et al., 1954*; *Beutler, 1959*) and later exploited by Alving et al. to develop the once weekly regimen in G6PDd (*Alving et al., 1960*.)

By experimenting with high-dose weekly regimens and low-dose daily regimens, Beutler and colleagues showed haemoglobin would first fall as a result of oxidant haemolysis and then rise despite continued exposure to the same doses of primaquine which had caused the initial haemolysis. This resulted from reactive erythropoiesis (reticulocytosis) that introduced a younger red cell population to the circulation which was essentially 'resistant' to the haemolytic effects of that primaquine dose. Intermittent primaquine administration resulted in progressively smaller cycles of haemolysis followed by reticulocytosis as the red cell population became younger. These results led to a recommendation for a high-dose, once weekly primaquine regimen for radical cure in vivax malaria patients with G6PDd (8 once weekly adult doses of 45 mg) (*Alving et al., 1960*). This regimen was devised based on studies in subjects with the African $A^-$ variant of G6PDd, which is one of the mildest deficiencies. Safety was not formally assessed in more severe deficiencies. A recent trial of this regimen in vivax malaria patients with the more severe Viangchan G6PDd variant from Cambodia showed a greater fall in haemoglobin and a delayed recovery from anaemia in G6PDd compared to G6PD normal patients with one patient requiring a blood transfusion (*Kheng et al., 2015*). These data suggest that weekly primaquine may not be the optimal regimen for the more severe G6PDd variants prevalent outside Africa.

Reconsideration of the detailed haematological studies that laid the foundation for the weekly regimen suggests that an ascending-dose regimen of primaquine, with a schedule that matches the dynamics of red blood cell production, could induce a safe 'slow burn' haemolysis, even in individuals with severe G6PDd variants, and would still deliver a total therapeutic dose for radical cure.

Accordingly, our study had two objectives; first, to construct a compartmental model for red blood cell dynamics which could be used to analyse all available data from past studies of haemolysis in G6PDd individuals, and second to predict an optimal ascending dose regimen which would be safe and efficacious yet practical and could, therefore, be recommended without G6PD testing.

## Results

### Model fit

*Figure 1* shows hypothetical data simulated from the compartmental model with a primaquine regimen of 45 mg weekly for eight weeks fitted to data from adult G6PD deficient Cambodian patients. Parameters were randomly drawn from the Bayesian posterior distribution.

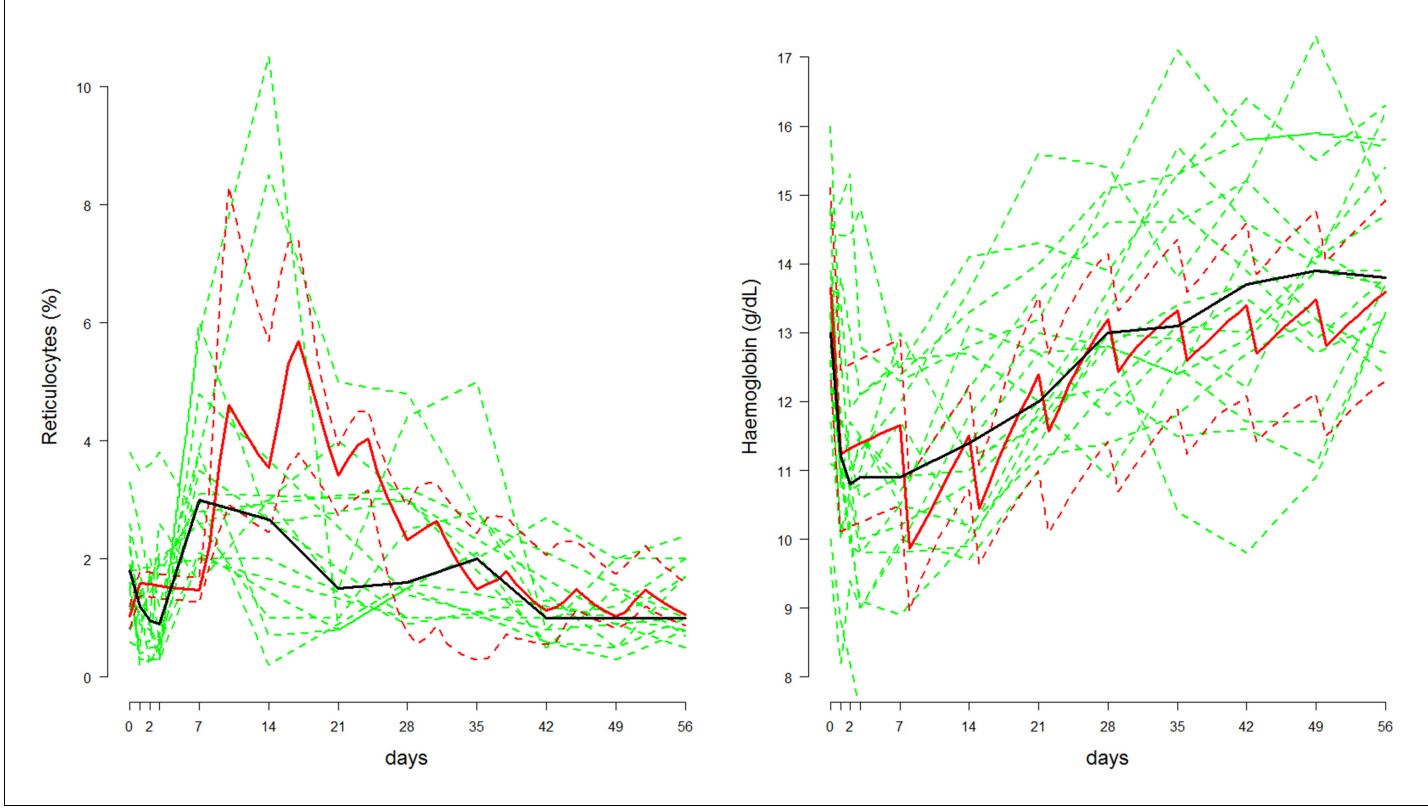

**Figure 1.** Comparison between the data from *Kheng et al. (2015)* (shown in green, population median in thick black line) and posterior predictive 80% credible intervals (shown in red, median: thick line; 10&90% boundaries: dashed lines) in which adult Cambodian patients who were G6PD deficient were given weekly primaquine (45 mg) for eight weeks. *Left*: reticulocyte response; *Right*: haemoglobin response.

The signal-to-noise ratio in the reticulocyte data is low and this is apparent from the median reticulocyte count which varies considerably during the 56 days. In comparison, simulations from the mechanistic model show that a substantial rise in the reticulocyte count should occur approximately one week after the first dose, with a peak after the third dose, and then return to normal slowly over the subsequent six weeks. The serial haemoglobin data on the other hand show a clear trend with a large fall after the first dose, a smaller fall after the second and then a gentle recovery with no major effect from subsequent primaquine doses. This trend is reproduced by the model and the posterior distribution also characterises satisfactorily the variance observed in steady state haemoglobin concentrations.

## Predicted dose response

Combining the data from *Figures 2–4*, it is possible to estimate a primaquine dose-haemoglobin response curve for G6PDd individuals whose severity is similar to the 'moderate severity' variants G6PDd Mahidol/Viangchan. The data at different dosing levels are sparse and the studies have been done in very different contexts; however, the strong mechanistic assumptions used to construct the compartmental model regularize the problem enough to compare the studies in a principled way. The data from G6PDd Mediterranean are excluded from this dose-response curve estimation because the haemolysis observed with this variant is considerably greater than for G6PDd Mahidol/Viangchan. However, the observed falls in haemoglobin after 5 daily doses of 30 mg in G6PDd Med Sardinians are shown by the red triangles in *Figure 5*, right plot, for comparison.

The posterior MCMC samples inferred from the Kheng data can be used to approximate model uncertainty around the median dose-response curve. The right plot of *Figure 5* shows the posterior predictive dose-response curve with 90% credible intervals, where the 'response' is defined as the drop in haemoglobin after five days at a given dosing level. Overlaid are estimates of the falls in

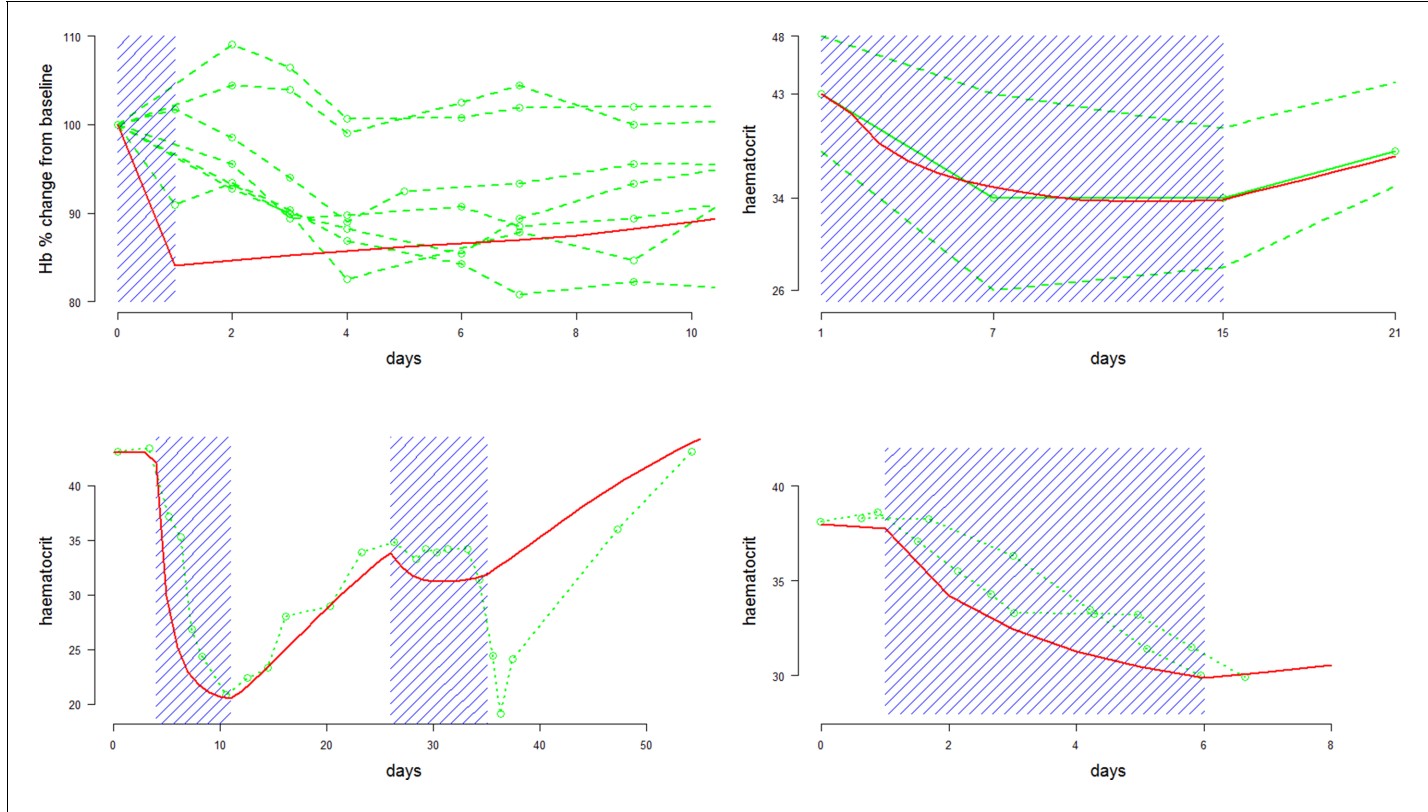

**Figure 2.** Comparison between approximate model fits (red) and data (green) extracted from four primaquine studies with a single dose or daily regimens all at 30/45 mg adult doses. Dosing periods are shaded in blue. The top two plots are for Mahidol and Viangchan variants, respectively. The bottom two plots are for the Mediterranean variant. From top left to bottom right: single 45 mg dose given to 7 G6PDd Mahidol Thais (*Charoenlarp et al., 1972*); 14 daily doses of 30 mg given to 15 G6PDd presumed Viangchan variant Khmer soldiers (only mean and extreme values reported) (*Everett et al., 1977*); 1 G6PDd Med Sardinian given two courses of daily 30 mg doses (*Pannacciulli et al., 1965*); 2 G6PDd Med Sardinians given 5 daily doses of 30 mg (*Salvidio et al., 1967*).

haemoglobin induced by 5 daily doses from studies in *Figures 2,3*, and an extrapolated estimate from the posterior distribution of the model fitted to data from weekly dosing in Viangchan variant.

It is of interest to compare the fitted dose-response relationship in *Figure 5* (right: thick black line)—corresponding to the more severe variants of G6PDd—with the green crosses corresponding to observed and fitted haemolysis in G6PDd African $A^-$ (mild variant). As would be expected, for the mild variant the dose-response relationship has the same shape but is shifted to the right.

## Safe optimal regimen

The currently recommended dose for the radical cure of vivax malaria in an adult in SE Asia and Oceania delivers 420 mg (that is, 30 mg/d x 14 d) of primaquine and is very effective (*John et al., 2012*). The maximum primaquine dose administered in the weekly regimen is 360 mg (8 x 45 mg) but the efficacy of this regimen has only been reported in Afghan refugees in Pakistan, a country with a relatively low relapse rate (*Leslie et al., 2008*).

The primary objective of our research is to design a novel primaquine regimen that could be given safely to individuals with G6PDd or of unknown status without G6PD testing and deliver a total dose that would be efficacious. The scientific hypothesis is that the same total dose could be given safely with tolerated declines in Hb over a longer duration by starting with a lower initial dose which is increased gradually over time. The ascending dose regimen would allow for a steady adjustment of the age distribution of RBCs by both slow primaquine-induced haemolysis and the resulting increased erythropoiesis. These results only concern ascending dose regimens given over 20 days. There are two reasons for this; first, adherence to long course regimens is likely to be poor, and

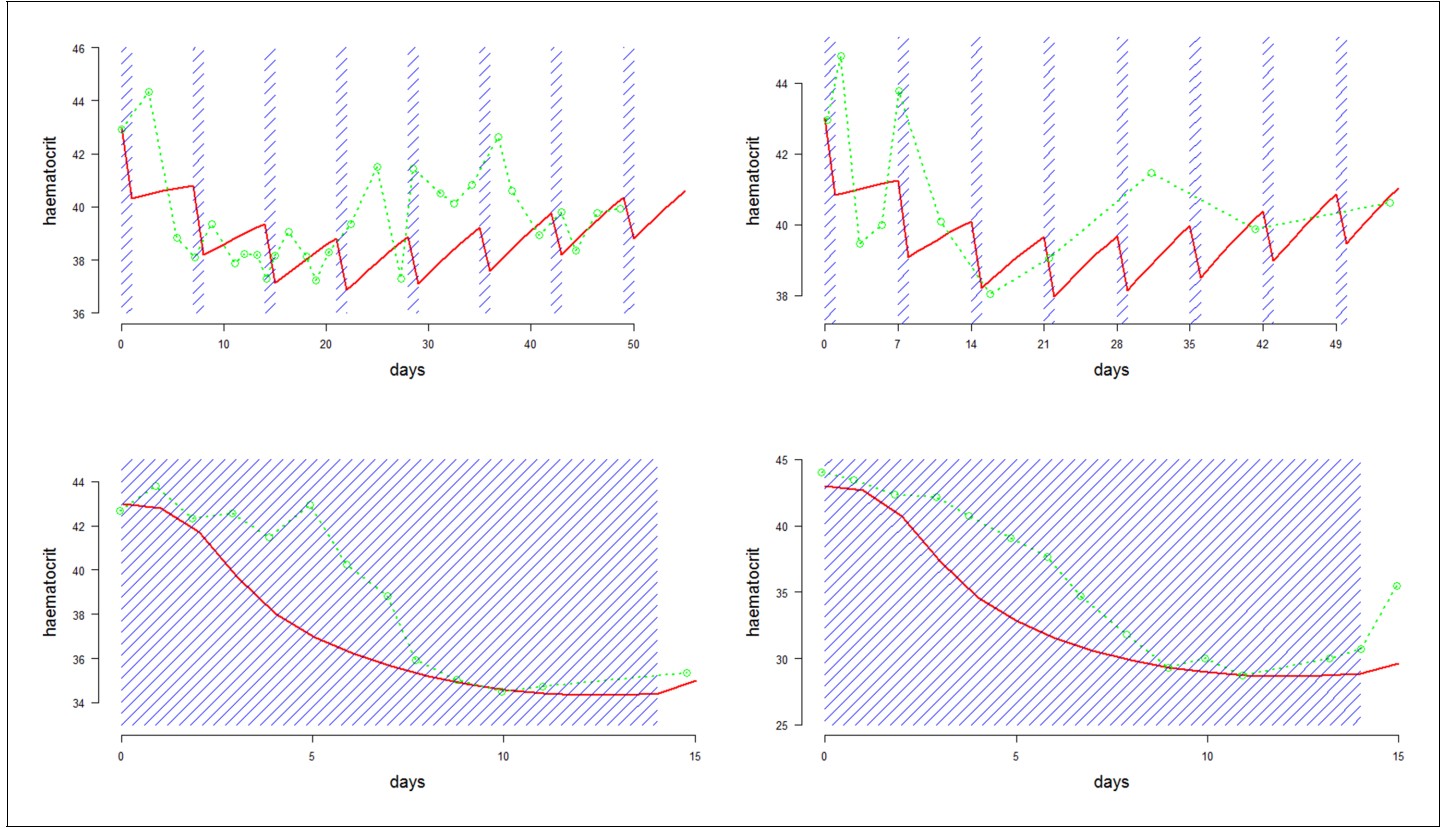

**Figure 3.** Comparison between approximate model fits (red) and data (green) extracted from four primaquine studies on the same individual with G6PDd African $A^-$(*Alving et al., 1960*). Dosing periods are shaded in blue. The top two plots are for weekly dosing regimens (8 doses): left is 60 mg per week; right is 45 mg per week; the bottom two plots are daily dosing regimens (14 doses): left is 15 mg per day; right is 30 mg per day.

second, the first relapses emerge from the liver about 14 days after starting treatment so the primaquine regimen has to provide sufficient drug to prevent the emergence or eliminate these parasites.

## Definition

For practical purposes, an acceptable ascending dose regimen is defined as a monotonic increasing dose regimen satisfying the following conditions: (i) the total dose is >380 mg , (current tablet sizes do not allow for a regimen to provide 420 mg easily to all adult patients—and 380 mg is considered to give similar efficacy); (ii) every increment is a multiple of 2.5 mg; (iii) the minimum adult daily dose is 5 mg; and (iv) the maximum adult daily dose is 30 mg.

The optimal ascending-dose regimen is defined as the one resulting in the slowest haemolysis, where the rate of haemolysis is penalized by the squared gradient. The optimisation problem is non-convex for all ascending dose regimens, so the solution is approximated using a greedy search algorithm. An estimated optimal dosing regimen satisfying the criteria defined above is shown in *Figure 6*, plotted in red (left: haemolytic effect; right: daily dosing of the ascending regimen). This was found using the median Bayesian posterior parameter estimates and a dose-response relationship taken from a linear interpolation of all points in *Figure 5* (left plot). In blue is a simplified version of this ascending-dose regimen, broken into four 5-day cycles at a fixed dose. The resulting haemolysis from the blue regimen is greater, and the drops in haemoglobin are more irregular (left plot). *Video 1* in the supplementary materials illustrates the red blood cell dynamics over the course of this regimen.

Although intuitively one might think that starting with a lower dose was safer, such a regimen can be in fact worse. This is shown in *Figure 5* which compares the haemolysis resulting from four regimens given in *Table 1*. Regimen D delivers too little primaquine at the start (observe very small

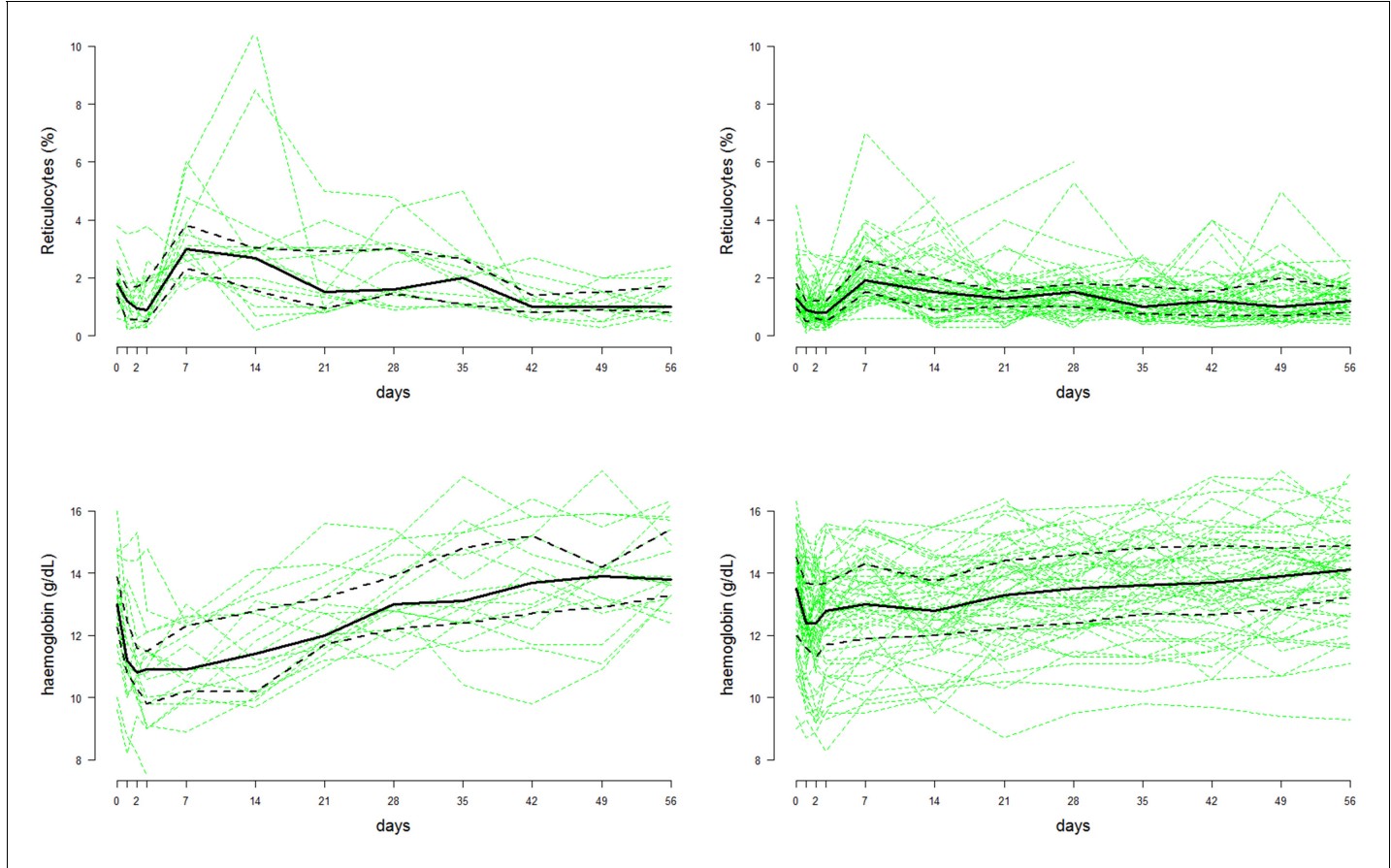

**Figure 4.** Time series data of reticulocyte count (top row) and haemoglobin concentrations (bottom row) from the Cambodian study on G6PDd individuals ($n = 18$, left column) and G6PD normals ($n = 57$, right column) (*Kheng et al., 2015*). The faint green lines show individual patient data; the thick black lines represent the population median values at each time-point; the dashed black lines show the interquartile range.

The following source data is available for figure 4:

**Source data 1.** This provides the source data for the reticulocyte counts and haemoglobin concentrations over time from the *Kheng et al. (2015)* study on weekly high-dose primaquine.

decreases in haemoglobin concentration) with a reticulocyte response that is too weak to render the RBCs 'resistant' to primaquine; the necessity to increase the PQ dose too fast to compensate for the slow start and to deliver an efficacious total dose results in a large drop in Hb on day 22.

## Discussion

Primaquine is widely recommended for the radical cure of vivax malaria but it is often not given because testing for G6PD deficiency is not widely available outside large centres. This has deleterious consequences for vivax malaria affected communities because it is the multiple relapses of vivax malaria from liver hypnozoites that cause substantial morbidity.

Seminal research conducted over 50 years ago characterized the biology of oxidant haemolysis caused by primaquine and provided an alternative once weekly regimen for patients who were G6PDd based on controlled haemolysis. This was shown to be safer in adult subjects with the 'mild' African $A^-$ variant of G6PDd, but was recommended for all G6PDd variants with variable adoption by countries since. In some countries (for example, Iran) it is the standard radical treatment for all patients. The safety and effectiveness of the high dose weekly regimen have been studied little over the past five decades.

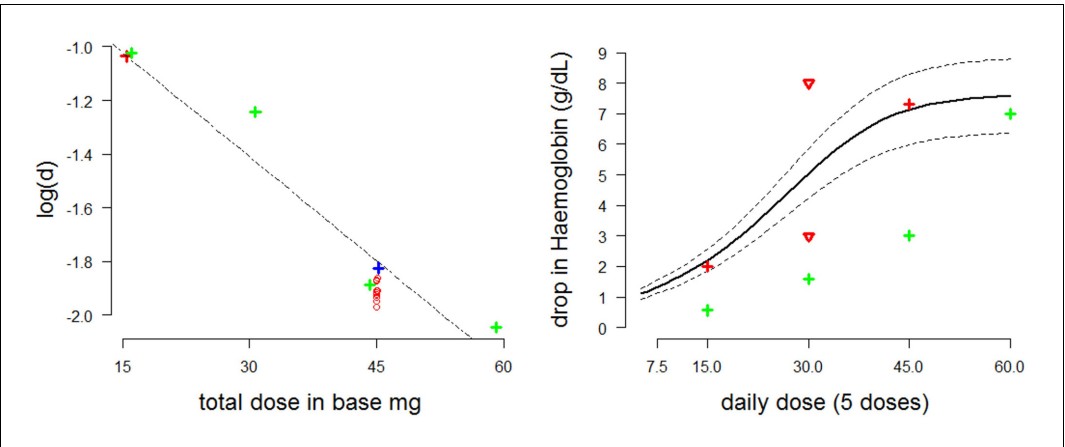

**Figure 5.** Estimating the dose-response curve for moderate/severe G6PDd. Left: estimates of the $\log d$ parameter as a function of the administered dose plotted with a linear regression curve (red cross: Viangchan; red circles: posterior estimates from model fitted to data from G6PDd Viangchan; blue cross: Mahidol; green crosses: African $A^-$). Right: dose-response curve (thick black line) with 90% credible intervals (dotted black lines) as measured by fall in haemoglobin (*y*-axis) after five days at a given dose (*x*-axis) based on draws from the posterior distribution. The red and green crosses are the estimated falls after five days from Viangchan and African $A^-$ studies, respectively (see *Figures 2,3*). The red triangles show the falls observed in G6PDd Med studies from *Figure 2*.

Uncomplicated malaria treatment recommendations are usually a trade-off between dosing precision and operational feasibility. A regimen which is long or complicated may be adhered to poorly. In this particular case it must also be able to prevent or suppress relapsing P. *vivax* or P. *ovale* parasites which begin to emerge from the liver as early as two weeks (becoming patent about one week later) in SE Asia and Oceania. This modelling exercise, based on all available data, sought to devise a primaquine regimen which would be safer in G6PDd patients, and, therefore, might be deployed without G6PD testing. It was calibrated against recent data in Cambodian patients most of whom

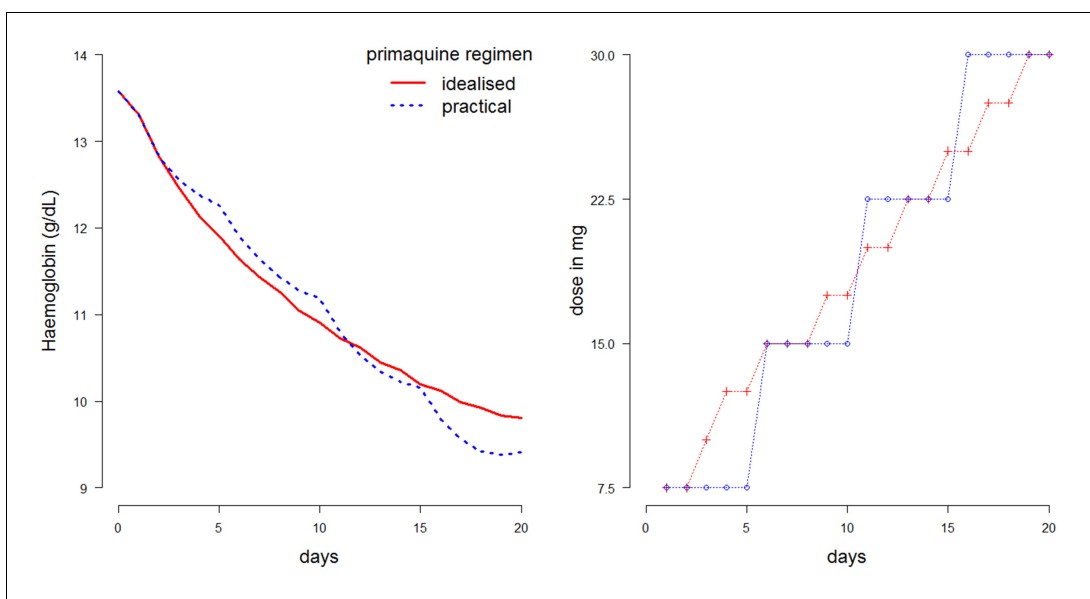

**Figure 6.** Comparison of two 20-day ascending-dose regimens. *Left*: haemolysis over time resulting from regimens. Blue: simplified regimen; red: idealized optimal regimen. *Right*: daily dosing construction for the two regimens. Total dose of blue regimen is 375 mg; total dose of red regimen is 382.5 mg.

had the Viangchan G6PDd variant. Thus, the model predictions of the degree of haemolysis and the tolerability and safety profile would be expected to hold for variants with similar or less severe enzyme abnormalities, but it would not necessarily hold for more severe variants such as G6PDd Mediterranean where more clinical research is required.

Under all circumstances, the ascending regimen proposed here would be expected to be safer than the current 14 day regimens in G6PDd hemizygous males and homozygous females, especially the 0.5 mg/kg regimen needed for frequent relapsing P. vivax. This is clinically relevant also for female heterozygotes. Even with current rapid testing methods (for example, fluorescent spot test and RDTs) which generally detect patients with $\leq 30\%$ normal G6PD activity, the haemolytic risk in heterozygote females, who may be classified erroneously as 'G6PD normal', could still be substantial. Up to $\approx 70\%$ of their erythrocytes may be G6PD deficient, and clinically significant haemolysis may result from daily higher dose primaquine regimens given to female heterozygotes (*Chu et al., 2017*).

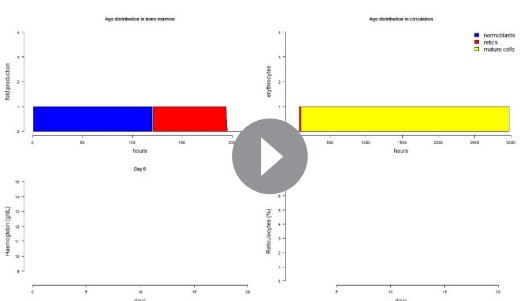

**Video 1.** Animated video showing the red blood cell dynamics for our optimal ascending dose regimen.

Although this compartmental model of RBC dynamics is highly simplified, it reproduces the essential dynamics of the body's response to primaquine-induced haemolysis in both healthy individuals and malaria patients. It can therefore help to guide the design of a Phase I study to evaluate its predictions, and thereby develop an optimal ascending dose regimen of PQ. An adaptive design protocol has been developed to test the simplified regimen (A) in G6PDd Mahidol healthy volunteers. A study in healthy G6PDd volunteers is essential to characterise the haemolytic response. Data from such a study can then be used to determine an optimal regimen which would then be tested for safety, and efficacy (that is, radical cure) in vivax malaria patients in a Phase II (that is, to define the PK-PD relationship in patients). Whether patients would adhere sufficiently to a longer regimen is an important operational concern so the optimised regimen would then need to be assessed for safety and effectiveness in larger field trials.

This use of mathematical modelling such as this could also be readily applied to the slowly eliminated 8-aminoquinoline tafenoquine, currently being tested for safety and efficacy in humans (*Beck et al., 2016*). Tafenoquine has the great advantage of being administered as a single dose for radical cure due to its long terminal elimination half-life. However, this means it could be dangerous in G6PD deficiency. Whereas the rapidly eliminated primaquine can be stopped if there is significant haemolysis, limiting the haemolytic effect, the haemolytic effect of the slowly eliminated tafenoquine cannot be readily reversed and so haemolysis will continue until all susceptible red cells are destroyed. Combined regimens for G6PDd patients in which primaquine is given initially to induce controlled haemolysis followed by tafenoquine might be possible, and would allow shorter total treatment durations. Data on the Hb response to different doses of tafenoquine would be necessary to calibrate the model.

**Table 1.** Illustrative regimens. A is our proposed optimal ascending dose regimen; B is a slight variation on this regimen (accelerated); C is a slower ascending dose regimen (potentially suitable for more severe variants); D illustrates a very poor regimen.

| Regimen | Day | | | | | |
|---|---|---|---|---|---|---|
| | 1-5 | 6-10 | 11-15 | 16-20 | 21-25 | 26-30 |
| A | 7.5 mg | 15 mg | 22.5 mg | 30 mg | - | - |
| B | 7.5 mg x3 d;10 mg x 2 d | 15 mg x 3 d; 17.5 mg x 2 d | 20 mg | 22.5 mg x 3 d; 25 mg x 2 d | - | - |
| C | 5 mg | 10 mg | 15 mg | 20 mg | 25 mg | - |
| D | 5 mg | 5 mg x 2 d; 10 mg x 3 d | 10 mg x 4 d; 15 mg x 1 d | 15 mg | 15 mg x 1d; 30 mg x 4d | 30 mg x 3 d |

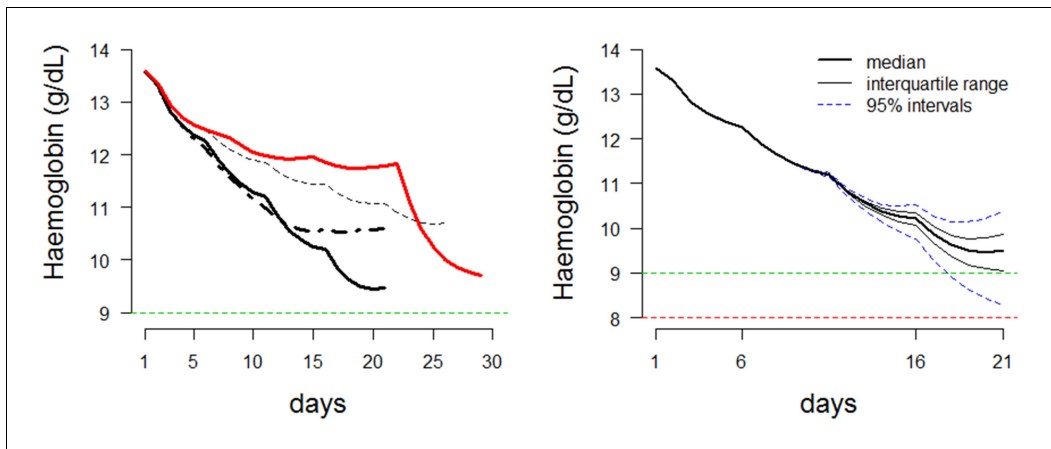

**Figure 7.** Dynamics of ascending regimens. *Left*: Comparing the haemolytic effect of four regimens. Thick black line: proposed optimal regimen; thick black dashed line: more conservative regimen with lower total dose; thin black dashed line: longer duration regimen for more severe variants; thick red line: bad choice regimen. *Right*: Posterior predictions for the proposed ascending dose for a given starting haemoglobin (steady state). Prediction using the median posterior values is shown by a thick black line. Predictions for 100 random draws from the posterior are shown by dashed blue lines. The horizontal line at a haemoglobin concentration of 9 is a proposed conservative 'safety threshold'. Horizontal line at a haemoglobin concentration of 8 is a proposed regimen limiting toxicity threshold.

The results of this study show how care will need to be taken when designing an ascending primaquine dose regimen in order to minimize falls in haemoglobin. This is shown by the toxic regimen D (*Table 1* and *Figure 7*). This gives an insight into the 'memory' property of the ascending dose regimens. The effect of a 30 mg dose will entirely depend on which doses were given during the previous days. In conclusion, these results suggest that an ascending PQ dosing regimen for vivax radical cure might be well tolerated and effective in mild or moderately severe G6PDd variants. These predictions should now be tested in an adaptive phase I study.

**Table 2.** Parameters and functions of the compartmental model along with their interpretation.

| Parameter | Units | Meaning |
|---|---|---|
| $d$ | unitless | Parameter of age-dependent killing function. |
| $Hb^*$ | Hb | Steady state haemoglobin concentration. |
| $Hb_t$ | Hb | Haemoglobin concentration at time $t$. |
| $\rho(Hb)$ | unitless | Fold-increase in production of RBCs as a function of haemoglobin concentration (at steady state $\rho(Hb^*) = 1$). |
| $\rho^{max}$ | unitless | Maximum fold increase in RBC production, this will be reduced in anemia. |
| $Hb_{50}^{\rho}$ | Hb | Haemoglobin concentration for which production is elevated to $\rho^{max}/2$. |
| $Release(Hb_t)$ | days | Time of release of reticulocytes into circulation as a function of haemoglobin concentration. |
| $k$ | unitless | Hill coefficient of sigmoid function $Release(Hb_t)$. |
| $Hb_{50}^{R}$ | Hb | Haemoglobin concentration corresponding to the mid-point of sigmoid describing reticulocyte release into circulation. |
| $T_{min}$ | hours | Earliest age of an RBC vulnerable to primaquine-induced haemolysis. |
| $T_{lag}$ | hours | Time to reach the maximum haemolytic effect of primaquine. |

## Materials and methods

### Mathematical model

The structure of the model of red cell dynamics is similar to the compartmental model developed by (*Savill et al., 2009*). RBC dynamics are simulated by tracking the age distribution of the red blood cells in hourly blocks. The homeostatic dynamics, which maintain the number of red blood cells or haematocrit at a steady state are straightforward. At steady state, approximately 0.83% of RBCs are replaced each day and 1% of RBCs in the circulation are reticulocytes. Severe acute anaemia has two consequences in the bone marrow. Reticulocytes are released into the circulating blood at an earlier age and with increased erythropoiesis normoblasts may be released into the circulation (reported as nucleated RBCs) (*Hillman, 1969*). Previous iron turnover studies in humans following phlebotomy suggest sigmoid relationships for both of these processes (*Hillman, 1969*).

### Compartmental model of RBC dynamics

The steady state haemoglobin concentration is denoted $\mathrm{Hb}^*$. The time at which reticulocytes are released into circulation is a function of the haemoglobin concentration at time t, denoted Release $(\mathrm{Hb_t})$. At steady state, when $\mathrm{Hb_t} = \mathrm{Hb}^*$, the reticulocytes mature in the bone marrow for 3.5 days (that is, Release $\{\mathrm{Hb}^*\} = 3.5$) and then spend approximately one day in the circulation before becoming erythrocytes. In anaemia, release can occur after only one day (then the cells are reticulocytes for 3.5 days in the circulation). These relationships were shown from plasma iron turnover studies following phlebotomy (*Hillman, 1969*).

By modelling the number of circulating RBCs directly, it is possible to compute the haemoglobin concentration at each time point in the simulations. The steady state number of RBCs corresponds to steady state haemoglobin and steady state haematocrit. The relationship between haematocrit and haemoglobin is assumed to be linear (*Lee et al., 2008*). At steady state the body produces approximately $10^8$ RBCs per hour. In the model, this is used as the baseline production quantity, represented by a production factor of $\rho = 1$. In extreme anaemia this can be increased fivefold or more, for example, $\rho \geq 5$. Both Release $(\mathrm{Hb_t})$ and $\rho(\mathrm{Hb_t})$ are modelled as sigmoid functions:

$$\rho(\mathrm{Hb}_t) = \frac{\rho^{\mathrm{max}}}{1 + e^{\lambda(\mathrm{Hb}^*)(\mathrm{Hb}_t - \mathrm{Hb}_{t_{50}}^\rho)}}$$

$$\mathrm{Release}(\mathrm{Hb}_t) = 1 + \frac{2.5}{1 + e^{-k(\mathrm{Hb}_t - \mathrm{Hb}_{t_{50}}^R)}}$$

with $\lambda(\mathrm{Hb}^*)$ given by:

$$\lambda(\mathrm{Hb}^*) = \frac{\log(\rho^{\mathrm{max}} - 1)}{\mathrm{Hb}^* - \mathrm{Hb}_{50}^\rho}$$

where $\mathrm{Hb}_t$ is the haemoglobin concentration at time $t$ (or equivalently the haematocrit); $\rho^{\mathrm{max}}$ is the maximum fold increase in steady state RBC production; $\mathrm{Hb}^*$ is the steady state haemoglobin concentration; $\mathrm{Hb}_{50}^\rho$ is the mid-point of the $\rho$ sigmoid function (the haemoglobin concentration at which production is equal to $\rho^{\mathrm{max}}/2$; $k$ the Hill coefficient which regulates the steepness in response to perturbations in haemoglobin levels; $\mathrm{Hb}_{50}^R$ the mid-point of the release function.

For simplicity, it is assumed that in the normal healthy state all red blood cells live exactly 120 days as erythrocytes, and therefore these two functions are sufficient to model the feedback loops which regulate perturbations to haemoglobin levels.

The following class of functions is used to model the red cell age-dependent primaquine-induced haemolysis:

$$Probability(\mathrm{PMQ\,induced\,death\,at\,age\,t}) = \begin{cases} e^{(t-120)^d} & t \in [T_{min}, 120] \\ 0 & t \leq T_{\min} \end{cases} \tag{1}$$

where $T_{\min}$ is the age of the youngest red blood cells lysed by primaquine. This parameter varies as a function of the degree of G6PD deficiency (determined by the genetic variant of G6PDd). The data from (*Pannacciulli et al., 1965*) suggest cells as young as 16 days can be lysed with a daily

primaquine dose of 30 mg in the severe Mediterranean variant, whereas in the less severe African A-variant haemolysis appears confined to cells older than 50 days (*Beutler et al., 1954a*). The steepness of this 'killing function' is regulated by the parameter $d$, with smaller values of $d$ giving a sharper drop in haemoglobin levels.

Depending on the degree of severity of G6PD deficiency, haemolysis will be observed more or less quickly after the first dose of primaquine. To simulate this effect, a time lag component is added to the model, the value of which will depend on the genetic variant of G6PDd. The time lag component, denoted $T_{lag}$, reduces the total haemolytic effect (as given in *Equation 1* for the first $T_{lag}$ hours after the first dose of primaquine. A glossary of all parameters of the model alongside their units and interpretation is given in *Table 2*.

Available data on primaquine induced haemolysis are sparse (these data are reviewed in detail in the next section). Thus, many of the free parameters such as $T_{lag}$ and $T_{min}$ are fixed using expert opinion and their impact on the modelling evaluated by a sensitivity analysis shown in appendix Modelling primaquine-induced haemolysis in G6PD deficiency. A primary goal of this analysis was to parameterize the relationship between primaquine dose and haemolysis for a given severity of G6PD deficiency. Thus the dose-response curve varied with the different genetic mutations. A preliminary analysis (Figure 5) of the available historical data (*Figure 2*) suggests that the relationship between dose and age-dependent haemolysis as parameterised by the function in *Equation 1* is logarithmic:

$$\log d = \alpha PMQ_{\text{dose}} + \beta x + c$$

where $x$ are individual covariates of importance such the G6PDd variant and sex (the deficiency is X-linked) and $PMQ_{\text{dose}}$ is the dose of primaquine.

## Inputs and outputs of model

The goal of this compartmental model was to simulate haemolysis following primaquine administration to a G6PDd individual over a fixed period of time. In order to do this, the model needs as inputs the following elements.

1. The age distribution of RBCs both in the bone marrow (normoblasts and reticulocytes) and in circulation (reticulocytes and erythrocytes). This is represented as a vector of counts of RBCs for each age group discretized into hourly blocks.
2. The number of hours for which to simulate the model forward in time. This includes a dosing schedule (binary vector discretized in hourly blocks where 1 represents drug in the body and 0 represents drug absent). In all the simulations shown, the drug schedules are designed in multiples of 24 hr blocks (the terminal elimination half-life of primaquine is $\approx 5$ hours). In this manner it is possible to simulate weekly dosing and daily dosing.
3. The dosing level of the drug (this is defined by the value of parameter $d$ from *Equation 1).*

The simulations output the haemoglobin concentration over time, the reticulocyte percentage over time, and the final age distribution of RBCs in both the bone marrow and in circulation.

The structure of the model also allows for the simulation of the effect of malaria within an individual by altering the age distribution as *P. vivax* invades young erythrocytes exclusively. It is assumed that healthy individuals with no history of haemolytic events in the last 4 months will have a uniform age distribution of RBCs. Simulations in *P. vivax* infections could be done by shifting the age distribution. Model code is available in *Source code 1*. *Supplementary file 1* provides posterior MCMC samples from the model.

## Data

### Historical studies

Although primaquine was first tested in humans in 1944 and approved by the US FDA in 1952, there are very few precise data on its haemolytic effect in G6PDd individuals. Most of the studies only present sparse data, often limited to summary statistics, and with small sample sizes ($n \approx 2$). This section presents an analysis of the most information-rich studies conducted over the last 60 years and shows how they can be used to design informative prior distributions for the compartmental model. An exhaustive review of all available data on the haemolytic effects of primaquine has been reported recently by *Recht et al. (2014)*. There are however only five studies which present useful data on

falls in haemoglobin concentrations over time. Throughout this section, for consistency in *Figures 1–4*, data are shown in green and the model fits/predictions in red.

The data extracted from past studies in healthy volunteers are shown in *Figures 2,3*. These figures compare least squares model fits with the data. The studies span four different variants of G6PD deficiency: Mediterranean, Mahidol, Viangchan (The Khmer soldiers whose data are shown in *Figure 2* are incorrectly referred to by *Everett et al. (1977)* as G6PDd Mahidol. They are in fact most likely to be G6PDd Viangchan, as the variant was discovered some ten years later [1988]) and African $A^-$. These can be categorised generally as severe, moderate, moderate and mild variants of G6PDd, respectively.

The deterministic compartmental model reproduces the essential patterns of the observed dynamics, namely, a more rapid decrease in haematocrit at the start of the regimen which then slows as the haemolysis becomes self-limiting. However, some aspects cannot be reproduced by the model. The bottom left plot of *Figure 2* shows a sharp drop in haematocrit after the second round of primaquine administration. This is surprising, although the fact that it is outside the administration period may be an error in plotting in the original paper. The compartmental model cannot reproduce such a marked second drop after an initial fall of over 50% from baseline. In *Figure 3*, for the bottom two plots, the model predicts a faster initial decrease than observed for a similar nadir drop.

These data were used to select a plausible range of values for the parameter $d$ in *Equation 1*. This parameter governs the age-dependent haemolysis at different dosing levels.

## Weekly high-dose primaquine in G6PDd viangchan

The largest and most recent study of primaquine-induced haemolysis in G6PD deficiency is from *Kheng et al. (2015)*. In this study, 75 Cambodian patients with vivax malaria were given primaquine 0.75 base mg/kg weekly for 8 weeks. Of these, 17 were G6PDd Viangchan (14 homozygous males and three heterozygous females) and 1 was a homozygous male with G6PDd Canton. Haemoglobin concentrations and reticulocyte counts were measured on days 0,1,2,3 and subsequently each week, before the next dose of primaquine. One patient had a marked drop of haemoglobin falling to 7.5 g/dL and required a blood transfusion. The data are shown in *Figure 4*. The variation in the measurement of haemoglobin was approximately 1 g/dL. For reticulocyte counts, the measurement error is much greater when done by microscopy (that is, counting per 100 red blood cells).

## Model fitting using MCMC

The historical (*Figures 2,3*) and Kheng (*Figure 4*) data were used to fit the compartmental model; the former were used to select suitable prior distributions for parameters. Bayesian model fitting via MCMC was then applied to the data from the weekly high-dose primaquine in Cambodia (*Kheng et al., 2015*). The likelihood of the parameters is defined by a deterministic simulation from the compartmental model for a given dosing regimen and assumes both the haemoglobin levels and reticulocyte counts are observed with Laplace distributed errors (assuming Laplace errors is equivalent to minimizing absolute deviation). A Bayesian hierarchical structure was used for the steady state haemoglobin and the maximum increase in the production of red blood cells. This makes the assumption that each patient in the study is characterized by an individual steady state haemoglobin concentration $Hb^*$ and a maximum production capacity $\rho^{max}$ drawn from a population distribution (normal distributions in both cases). Weekly informative priors were used for all parameters and the posterior distribution was estimated using MCMC with a Metropolis-Hastings proposal. Details of prior distributions and histograms of posterior distributions, together with convergence diagnostics and summary statistics are in the Appendix 1, 'Structure of hierarchical model and MCMC diagnostics'.

## Acknowledgements

This study was part of the Wellcome Trust Mahidol University Oxford Tropical Medicine Research Programme funded by the Wellcome Trust. We thank Nick Savill for sharing his code. We thank Lisa White and Ben Cooper for help with designing the model.

# Additional information

## Funding

| Funder | Author |
|--------|--------|
| Wellcome | James Watson<br>Walter RJ Taylor<br>Nicholas J White |

The funders had no role in study design, data collection and interpretation, or the decision to submit the work for publication.

## Author contributions

JW, Conceptualization, Software, Formal analysis, Validation, Visualization, Methodology, Writing—original draft, Project administration, Writing—review and editing; WRJT, Data curation, Supervision, Investigation, Writing—review and editing; DM, SK, Data curation, Writing—review and editing; NJW, Conceptualization, Supervision, Funding acquisition, Validation, Investigation, Writing—review and editing

## Author ORCIDs

James Watson, http://orcid.org/0000-0001-5524-0325
Didier Menard, http://orcid.org/0000-0003-1357-4495
Nicholas J White, http://orcid.org/0000-0002-1897-1978

# Additional files

## Supplementary files

• Supplementary file 1. Posterior MCMC samples from model run on data from *Kheng et al. (2015)*.

• Source code 1. Model code.

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

## Appendix 1

# Structure of hierarchical model and MCMC diagnostics

The analysis from section 'Model fitting' assumes that the parameters $d, k, \text{Hb}_{50}^{\rho}, \text{Hb}_{50}^{R}$ are defined on a population level (same for all individuals in the study). The parameters $\text{Hb}^{*}, \rho^{\max}$ are defined on the individual level but drawn from a population distribution. Thus the $i^{th}$ patient is characterized by:

$$\text{Hb}_i^{*} \sim N(\theta_{\text{Hb}}, \sigma_{\text{Hb}}^2)$$

$$\rho_i^{\max} \sim N(\theta_{\rho}, \sigma_{\rho}^2)$$

We set informative priors on the hyperparameters $\theta_{\text{Hb}}, \theta_{\rho}$, namely:

$$\theta_{\text{Hb}} \sim N(40, 3^2)$$

$$\theta_{\rho} \sim N_{trunc}(4, 1; \min = 2, \max = 8)$$

where $N_{trunc}$ is a truncated normal distribution with upper and lower values given by $\min$ and $\max$. The variance hyperparameters $\sigma_{\text{Hb}}^2$ and $\sigma_{\rho}^2$ are given flat priors. **Figure 2** shows the posterior distributions for all the parameters and hyperparameters of the model. The population level parameters are given the following informative priors:

$$d \sim Beta(2, 38)$$

$$\text{Hb}_{50}^{\rho} \sim N(30, 2^2)$$

$$\text{Hb}_{50}^{R} \sim N(30, 2^2)$$

$$k \sim N_{trunc}(10, 2^2; \min = 0, \max = 20)$$

Convergence of the Metropolis Hastings algorithm was done by running four independent chains and computing the Gelman-Rubin statistic.

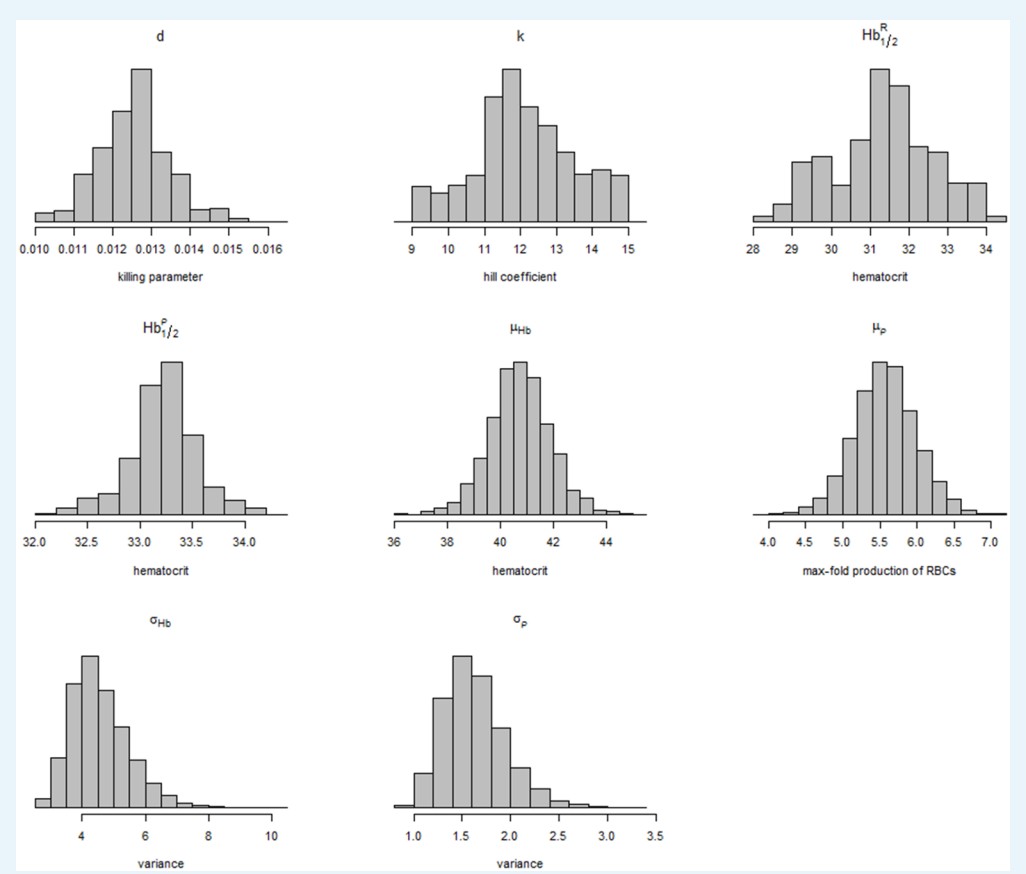

**Appendix 1—figure 1.** Posterior distributions of model parameters and hyperparameters.

As an independent check for model fit, *Figure 3*, compares the estimated individual steady state haematocrits with the data on individual mean corpuscular volume (MCV) on day 0. The relationship between these fitted estimates and the data from the study is approximately linear for all patients (G6PDd and G6PD normal). The MCV is a poor marker of anaemia; high values are associated with reticulocytosis, folate and B12 deficiency and low values with iron deficiency, and thalassaemia both of which are very common in tropical areas. A correlation with steady state haematocrit would suggest that the model is indeed estimating the correct quantities. Therefore if the model estimates of the steady state haematocrit are correct, this should in theory correlate well with the baseline MCV and suggest the model is converging to the correct quantities.

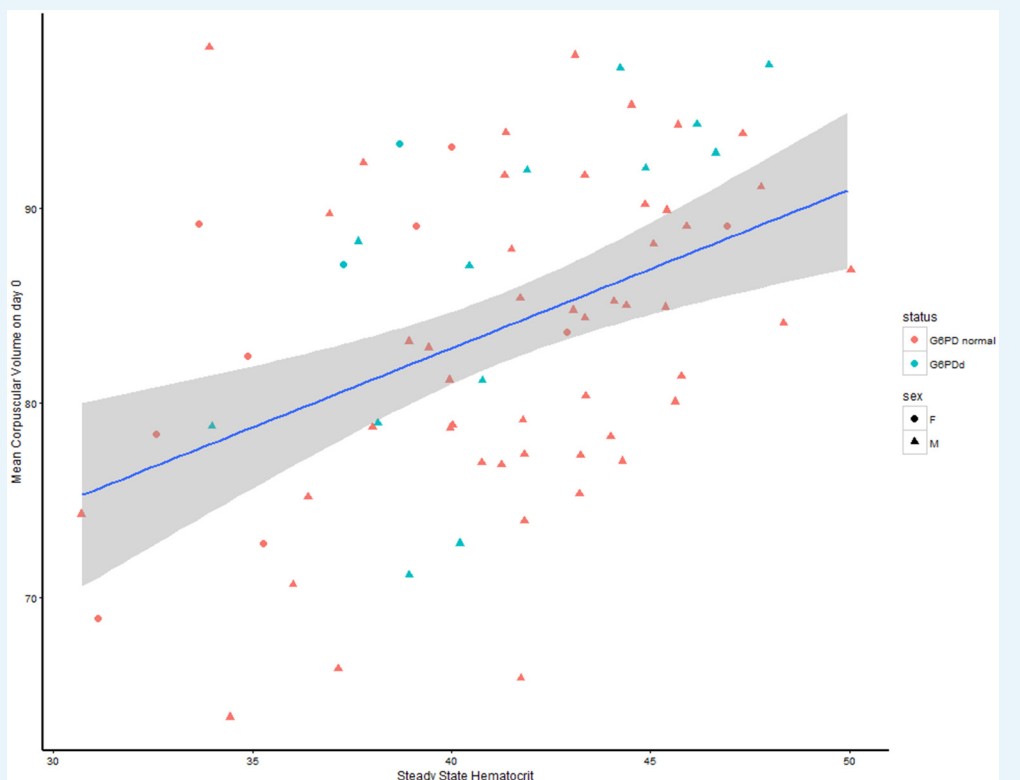

**Appendix 1—figure 2.** Relationship between the steady state haematocrit and the mean corpuscular volume at day zero. G6PDd patients are colored in blue, G6PD normal patients in red; women are shown by circles, men by triangles.

## Sensitivity analysis

*Figure 4* shows the effect of the individual parameters of the model (except the killing parameter $d$, the main parameter if interest in the model) on the haematocrit and reticulocyte response when all other parameters are held fixed.

To illustrate the individual effects of the parameters, we take as example a daily dosing regimen with identical dose (that is, same value of the parameter $d$) for 30 days. This is chosen to approximate the drop in haemoglobin observed in G6PDd African $A^-$ studies (*Figure 2*). We fix $d = 0.05$.

Some notable points:

- The max-fold increase ($\rho^{\max}$) of RBCs in the bone marrow has the most impact on the reticulocyte response after day 9. We assume that a twofold increase in production is the minimum viable response to anaemia (shown by the red lines). *Hillman (1969)* estimates the max-fold increase in healthy males as approximately 5-fold.
- The analysis of the data from *Kheng et al. (2015)* gives very similar values for the population distribution of $\mathrm{Hb}^\rho_{50}$ as that estimated by *Hillman (1969)* (see *Figure 2* in *Hillman [1969]*).

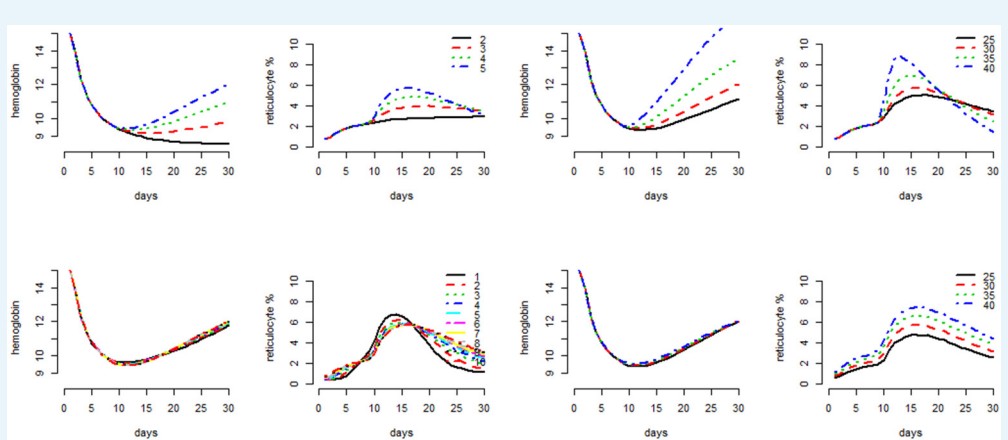

**Appendix 1—figure 3.** Individual parameter effects on the behavior of the compartmental model as shown by the haematocrit response and the reticulocyte count response. From top left to bottom right, grouped by pairs: the mid-haemoglobin concentration parameter $Hb_{50}^{R}$ for the reticulocyte release function; the mid-haemoglobin concentration parameter $Hb_{50}^{\rho}$ for the marrow production function; the hill coefficient $k$ for the reticulocyte release function; the max-fold production factor $\rho^{\max}$. The different values plotted for each parameter are shown in the legend for reticulocyte response plot.

