## [Decision Letter]

Thank you for submitting your article "Modelling primaquine-induced haemolysis in G6PD deficiency" for consideration by *eLife*. Your article has been reviewed by three peer reviewers, and the evaluation has been overseen by a Reviewing Editor and Prabhat Jha as the Senior Editor. The following individual involved in review of your submission has agreed to reveal his identity: Dennis Shanks (Reviewer #2).

The reviewers have discussed the reviews with one another and the Senior Editor has drafted this decision to help you prepare a revised submission.

Summary:

The paper needs to cite all available data and not the selected sources currently used. Some specifications about the genotypes and regimes need to be improved.

Essential revisions:

Reviewer #1

I've taken a quick look at the paper. It's not an area I know about, but the basic biology seems to have been described fairly clearly. The modelling is interesting and seems convincing but is a bit short on details. In particular I'd like to know:

a) More evidence that the five studies identified really capture the entirety of the useful literature – could they point to a recent literature review or even a commentary (or do a rapid review of their own)?

b) More information about the model fit/validation in Figure 2,Figure 3, particularly since the data represent individuals with a range of G6PDd variants. What algorithm did they use for the fit (they mention least squares but that is just a vague term for a large class of fitting methods), what parameters were used (and what produced by the fit), and do the resulting parameters correspond to what we know about genetic variation (and indeed geographic variation in treatment etc.)?

c) Given the variation in the historical studies, in what way were they used to generate priors for the MCMC algorithm? (It would be useful for the prior and posterior distributions to be shown, including the joint distributions and measures of association for the posteriors)

d) For the results, it would be more useful to present them in probabilistic terms rather than show multiple lines on a graph (and in some cases ignore uncertainty entirely).

Reviewer #2

The manuscript reviewed concerns mathematical modeling of an important severe adverse event seen during vivax malaria treatment with primaquine. Unlike most modeling papers this manuscript has a firm, practical conclusion in the form of an alternative dosing regimen. I also found the paper much more accessible than other modeling papers however I would state that I am not a modeling expert and the editor should have the manuscript reviewed by someone with such expertise. The numbers match clinical reality as I perceive it, but someone needs to check how the figures were generated. The conclusion that the proposed regimen needs to be tested in a clinical trial is rational and fully supported. It seems likely that the large Mahidol Oxford group is already doing so and if a clinical trial has been initiated something to that effect should be inserted into the Discussion.

If possible, it is important to ground a model in reality which is done by using all the available data. The authors do so and the sparse data is rather a comment on how little has been done recently on primaquine. This model would not have been possible without the same group's recent weekly primaquine study (Kheng et al., 2015) done in Cambodia. The best available estimates from mass drug administration programs in China in the 1970s and returning US military in the 1950s suggest that the Severe Adverse Event (massive hemolysis) rate is about 1:10,000. Could the authors insert these historical studies into their model to determine if this estimated rate matches what the model generates using 22.5 mg daily for 10 days (China) and 15mg daily for 14 days (US military)? This is certainly not essential but would be a good addition if it was in the realm of statistical possibilities.

If the suggested revised primaquine regimen passes a clinical trial, then this particular modeling exercise will be exceptional in having delivered an actual advance in public health practice.

Reviewer #3

This paper addresses a pertinent limitation to the treatment of *Plasmodium vivax* malaria and proposes a novel treatment regimen for an existing licenced drug. The analysis revisits a largely forgotten idea previously suggested in the 1950s which could revolutionise how *P. vivax* radical cure is administered and potentially greatly increase access to treatment and hence reduce the burden of *P. vivax* malaria.

The paper presents a model framework which utilises the available historical data about the dynamics of primaquine-induced haemolysis in G6PD deficient patients. The parameterised model is then applied to simulate the haemolytic impact of different primaquine treatment regimens on G6PDd patients and concludes with a recommended optimal regimen which in most cases would not cross the limiting toxicity threshold (assuming the parameters defined here). The paper is clearly and logically presented, though I am not a modeller so may have missed some technical aspects of the model. I have a few remarks:

1) Cambodia dataset genotypes: Specify the genotypes of the G6PDd patients from the Cambodian study. If any of these were heterozygous it would mask the full impact of the haemolysis on hemi- or homozygotes. Heterozygotes should be excluded from the model fitting to ensure that the resulting model is a "worse case" model for individuals with 100% G6PDd RBC.

2) Anaemia: The recommended "regimen A" triggers an overall drop in Hb of 4g/dL. While this is a progressive drop, it is nevertheless considerable. The starting Hb* (steady state Hb conc.) has important implications for the safety of this haemolysis, and anaemia will be common among *P. vivax* patients. The Hb* conc. used in Figure 7 and Figure 8 is the median value from the Kheng dataset, but Figure 4 shows there is wide variation around this. How are the Hb dynamics of the ascending regimens affected by a lower Hb*? Do the dynamics of the Hb drop differ or is it just a shift down the y-axis? The proposed "safety threshold" would be crossed if the starting Hb were 13g/dL or lower (which would be common), and an Hb*of <12g/dL could fall below the "limiting toxicity threshold". The issue of anaemia needs to be better addressed regarding the regimen's suitability for roll-out as a standard treatment.

3) The recommended regimen A has a total dose of only 375mg, while the opening statement of the paragraph "Safe optimal regimen" indicates that 420mg are necessary for radical cure. Further justification needed regarding the rationale for this discrepancy.

4) Aside from fears around drug-induced haemolysis, a major limitation to the current 14 day regimen is compliance. The proposed regimen A is 50% longer at 21 days. While this issue is beyond the scope of this paper, it should at least be acknowledged in the Discussion.

---

## [Author Response]

Essential revisions:

Reviewer #1

I've taken a quick look at the paper. It's not an area I know about, but the basic biology seems to have been described fairly clearly. The modelling is interesting and seems convincing but is a bit short on details. In particular I'd like to know:

a) More evidence that the five studies identified really capture the entirety of the useful literature – could they point to a recent literature review or even a commentary (or do a rapid review of their own)?

The most recent and most exhaustive literature review is the WHO report by Recht et al. We have added a comment and explain why we only choose these 5 studies (section Historical studies).

*b) More information about the model fit/validation in Figure 2,Figure 3, particularly since the data represent individuals with a range of G6PDd variants. What algorithm did they use for the fit (they mention least squares but that is just a vague term for a large class of fitting methods), what parameters were used (and what produced by the fit), and do the resulting parameters correspond to what we know about genetic variation (and indeed geographic variation in treatment etc.)?*

The model fits shown in Figure 2,Figure 3 are mainly illustrative of the Hb dynamics with primaquine-induced haemolysis and show that the model can reproduce these dynamics. We show the model maximum likelihood fits under Gaussian error for reticulocyte counts and Hb values (parameter values which minimize the squared distance between the predicted and observed time series). The likelihood surface is however very flat around these maxima due to the extremely small sample sizes. This is why we don’t report parameter values as they are not particularly meaningful or robust, but they can provide some idea of how to choose prior distributions (see next comment).

*c) Given the variation in the historical studies, in what way were they used to generate priors for the MCMC algorithm? (It would be useful for the prior and posterior distributions to be shown, including the joint distributions and measures of association for the posteriors)*

The point estimates found from the historical studies were used to centre weakly informative prior distributions. The appendix “*Structure of hierarchical model and MCMC diagnostics*” has both the exact prior distributions, and histogram plots of the posterior distributions (Appendix 1 Figure 2 in revised draft of paper).

*d) For the results, it would be more useful to present them in probabilistic terms rather than show multiple lines on a graph (and in some cases ignore uncertainty entirely).*

We agree that for Figure 1,Figure 5,Figure 7 it is more elegant to present the uncertainty in form of predictive credible intervals. This has been changed. However for Figure 7 (left plot) and Figure 6 (left plot) we have only plotted the median Hb decline for simplicity, otherwise the plots would be too cluttered. Owing to the lack of data and possible model misspecification the most important output of the analysis is a regimen which causes the least median Hb decline. Only clinical studies using this regimen could correctly characterize the variability of Hb decline in G6PD deficient individuals and thus inform as to its suitability (larger Phase II/III type study).

Reviewer #2

The manuscript reviewed concerns mathematical modeling of an important severe adverse event seen during vivax malaria treatment with primaquine. Unlike most modeling papers this manuscript has a firm, practical conclusion in the form of an alternative dosing regimen. I also found the paper much more accessible than other modeling papers however I would state that I am not a modeling expert and the editor should have the manuscript reviewed by someone with such expertise. The numbers match clinical reality as I perceive it, but someone needs to check how the figures were generated. The conclusion that the proposed regimen needs to be tested in a clinical trial is rational and fully supported. It seems likely that the large Mahidol Oxford group is already doing so and if a clinical trial has been initiated something to that effect should be inserted into the Discussion.

Mahidol Oxford Research Unit is indeed planning such a trial in human normal volunteers first, using an adaptive design based on the modelling results presented in this paper. The protocol is currently under submission to the ethics committee. We have added this to the Discussion.

If possible, it is important to ground a model in reality which is done by using all the available data. The authors do so and the sparse data is rather a comment on how little has been done recently on primaquine. This model would not have been possible without the same group's recent weekly primaquine study (Kheng et al., 2015) done in Cambodia. The best available estimates from mass drug administration programs in China in the 1970s and returning US military in the 1950s suggest that the Severe Adverse Event (massive hemolysis) rate is about 1:10,000. Could the authors insert these historical studies into their model to determine if this estimated rate matches what the model generates using 22.5 mg daily for 10 days (China) and 15mg daily for 14 days (US military)? This is certainly not essential but would be a good addition if it was in the realm of statistical possibilities.

These data are indeed interesting, although they may be underestimating the true severe adverse event rate because people rightly stop taking primaquine when they feel ill or pass black urine. Data from the Chinese MDA are summarized in a paper (Mass drug administration for the control and elimination of *Plasmodium vivax* malaria: an ecological study from Jiangsu province, China. Mal J 2013) but there are insufficient Hb concentration data in order to integrate them into our model.

The available Hb data, although sparse, inform the model as to the average trends in Hb decline; however, we do not feel confident predicting tail events (such as the probability of a severe adverse event) due to problems of model misspecification and lack of data.

Reviewer #3

[…]

1) Cambodia dataset genotypes: Specify the genotypes of the G6PDd patients from the Cambodian study. If any of these were heterozygous it would mask the full impact of the haemolysis on hemi- or homozygotes. Heterozygotes should be excluded from the model fitting to ensure that the resulting model is a "worse case" model for individuals with 100% G6PDd RBC.

The genotypes of the G6PDd patients from Kheng et al. are specified. There were a total of 18 patients who were G6PDd: 17 had the Viangchan variant (14 hemizygous males, 3 heterozygous females), and one male had the Canton variant.

We included the three heterozygote females as they had low and overlapping enzyme values compared to the homozygous males and therefore would not comprise our `worse-case predictions’.

2) Anaemia: The recommended "regimen A" triggers an overall drop in Hb of 4g/dL. While this is a progressive drop, it is nevertheless considerable. The starting Hb* (steady state Hb conc.) has important implications for the safety of this haemolysis, and anaemia will be common among P. vivax patients. The Hb* conc. used in Figure 7 and Figure 8 is the median value from the Kheng dataset, but Figure 4 shows there is wide variation around this. How are the Hb dynamics of the ascending regimens affected by a lower Hb*? Do the dynamics of the Hb drop differ or is it just a shift down the y-axis? The proposed "safety threshold" would be crossed if the starting Hb were 13g/dL or lower (which would be common), and an Hb*of <12g/dL could fall below the "limiting toxicity threshold". The issue of anaemia needs to be better addressed regarding the regimen's suitability for roll-out as a standard treatment.

Given the construction of the model, lowering the value of the steady state haemoglobin will lower the absolute drop but not the proportion of steady state drop (approx. 30% over 20 days). Therefore, a person with a steady state of 11 would drop to 7.7 g/dL, just under our proposed safety threshold. This Hb concentration would not be life-threatening and should rise promptly once primaquine has stopped.

Quantifying the true variation around these drops, especially in patients, is impossible given the lack of data and can only be done via correctly designed prospective clinical trials. We propose to test our proposed regimen first in healthy volunteers with sufficiently high Hb concentrations, i.e. 12 and above for a Phase I type study) and if it proves to be safe we will test the regimen in vivax patients to characterise the range of possible falls.

3) The recommended regimen A has a total dose of only 375mg, while the opening statement of the paragraph "Safe optimal regimen" indicates that 420mg are necessary for radical cure. Further justification needed regarding the rationale for this discrepancy.

These total doses were proposed to accommodate current tablet sizes and avoid tablet splitting. The relationship between total dose and anti-relapse efficacy has not been well characterised – there is very weak evidence for the superiority of the 30mg daily (total 420mg) versus 22.5mg (total 315mg) daily regimens in SE Asia. Elsewhere the original studies of the 45mg once weekly regimen (total 360mg) were conducted in African Americans who are on average 50% heavier than vivax malaria patients in SE Asia. One small trial in Afghan refugees in Pakistan (Leslie et al., 2008) showed that the weekly dose (360 mg total dose) was highly effective. Although not directly applicable to the more frequent relapsing forms of *P. vivax* in SE Asia, these data offer some reassurance. We think it very unlikely that efficacy of a 375mg regimen (~10% less primaquine) would be significantly lower than of a 420mg dose, and in this setting it is safety that will be a more important determinant of dosing than efficacy.

4) Aside from fears around drug-induced haemolysis, a major limitation to the current 14 day regimen is compliance. The proposed regimen A is 50% longer at 21 days. While this issue is beyond the scope of this paper, it should at least be acknowledged in the Discussion.

We have added a comment in the Discussion.